# The Impact of the COVID-19 Pandemic on Psychological Health and Insomnia among People with Chronic Diseases

**DOI:** 10.3390/jcm10061206

**Published:** 2021-03-14

**Authors:** Paweł Wańkowicz, Aleksandra Szylińska, Iwona Rotter

**Affiliations:** Department of Medical Rehabilitation and Clinical Physiotherapy, Pomeranian Medical University in Szczecin, Żolnierska 48, 71-210 Szczecin, Poland; aleksandra.szylinska@gmail.com (A.S.); iwrot@wp.pl (I.R.)

**Keywords:** COVID-19, chronic diseases, Hashimoto’s disease, GAD-7, PHQ-9, ISI

## Abstract

The outbreak of the severe acute respiratory syndrome coronavirus 2 (SARS-CoV-2) pandemic highlighted the serious problems of health care systems but also threatened the mental and physical health of patients worldwide. The goal of this study was to assess psychological health and insomnia in people with chronic diseases in the time of elevated stress associated with the pandemic. The study involved 879 people from Zachodniopomorskie province in Poland. Each participant provided basic demographic data, data on symptoms of insomnia, depression, anxiety and information on concomitant diseases such as hypertension, diabetes mellitus, coronary heart disease, heart failure, dyslipidemia, chronic obstructive pulmonary disease, Hashimoto’s disease and smoking cigarettes. Chronic diseases included in this study showed a strong correlation between Hashimoto’s disease and increase scores according to the Insomnia Severity Index (ISI, r = 0.797, *p* < 0.001), the Generalized Anxiety Disorder scale (GAD-7, r = 0.766, *p* < 0.001) and the Patient Health Questionnaire (PHQ-9, r = 0.767, *p* < 0.001). After the results were corrected for age, gender, diagnosed hypertension, dyslipidemia and cigarette smoking, it was confirmed that the diagnosis of Hashimoto’s disease was associated with an increased risk of anxiety (odds ratio (OR) = 2.225; *p* < 0.001), depression (OR = 2.518; *p* < 0.001) and insomnia (OR = 3.530; *p* < 0.001). Our study showed that during the SARS-CoV-2 pandemic patients with Hashimoto’s disease show a higher risk of insomnia, anxiety and depression.

## 1. Introduction

In 2020, Poland was forced by the severe acute respiratory syndrome coronavirus 2 (SARS-CoV-2) pandemic to introduce its first-ever national lockdown. The disease, commonly referred to as Coronavirus Disease 2019 (COVID-19), was first observed at the end of 2019, while the first case of infection in Poland was recorded on 4 March 2020. A much more infectious SARS, and with a higher mortality rate than influenza, COVID-19 has resulted in more than 84 million confirmed cases and 1.8 million deaths worldwide [1]. In Poland, novel coronavirus has killed almost 28,000 people and more than 1.3 million have been infected.

During previous coronaviral epidemics, such as SARS and Middle East respiratory syndrome (MERS), symptoms were more severe in people with chronic diseases and were often associated with a poor prognosis [2]. The Centers for Disease Control and Prevention (CDC) continually updates the list of conditions that increase the risk of undergoing a severe course of COVID-19. This list is not definitive and currently only includes conditions for which there is adequate clinical evidence. It allows clinicians to identify the most at-risk groups of patients for individualized prevention, treatment or care strategies. Chronic diseases listed by the CDC as those that can lead to a severe course of COVID-19 include cancer, chronic kidney disease, chronic obstructive pulmonary disease, Down’s syndrome, heart disease such as heart failure, coronary artery disease or cardiomyopathy, obesity, sickle cell anemia, smoking, diabetes, immune-weakening conditions due to blood or bone marrow transplantation, immune deficiencies such as in human immunodeficiency virus (HIV) and the use of corticosteroids or other immune-weakening drugs e.g., those for autoimmune diseases [3]. Autoimmune diseases, apart from cardiovascular diseases and cancer, are an increasing problem for the Western world. However, their exact causes are largely unknown and, moreover, are difficult to identify at the preclinical stage. It is likely that for this reason, they are not the most attractive topic for research in many countries, including Poland. Associated with abnormal functioning of the immune system, autoimmune diseases involve an undirected response against the body’s own organs, tissues and cells [4]. Currently, more than 20 million people suffer from autoimmune diseases in the United States alone, while epidemiological data provide evidence of a steady increase in their incidence in Western societies over the past few decades. Incidence rates of autoimmune diseases vary from less than 5 per 100,000 for uveitis to over 350 per 100,000 for Hashimoto’s disease [5]. Hashimoto’s disease is one of the most common autoimmune diseases in the population. It is difficult to diagnose and diagnosis often occurs in the later stages of the disease. The most common biochemical abnormalities in Hashimoto’s disease are elevated levels of thyrotropic hormone and low levels of free thyroxine combined with increased levels of antithyroid peroxidase antibodies. However, in the early stages of the disease, patients may be asymptomatic and biochemical test results may be normal which is associated with the periodic destruction of gland cells in this phase. This disease is most likely to affect young, active women [6]. Knowledge regarding autoimmune diseases appears to have a key role in the era of COVID-19. As the COVID-19 pandemic has progressed, there have been various reports on the development of autoimmune diseases in individuals infected with SARS-CoV-2. The mere entry of the SARS-CoV-2 virus into respiratory epithelial cells induces a severe inflammatory state known as a cytokine storm in some patients. Symptoms associated with autoimmune diseases such as fatigue, joint pain, muscle aches, and brain fog may appear during SARS-CoV-2 infection and persist even after infection. As well as this, autoimmune phenomena are supported by the presence of antinuclear antibodies, anti-DNA, or complement consumption in these patients [7]. In a study conducted by Bastard et al., it was observed that at least 101 of 987 patients with a life-threatening course of COVID-19 had neutralizing immunoglobulin G autoantibodies against interferon-ω, against interferon-α, or against both in the early stages of COVID-19. Several patients also had autoantibodies against the other three type I interferons [8].

The SARS-CoV-2 pandemic has seriously affected many aspect of human life and has dramatically threatened the mental and physical health of the general public. Ubiquitous information in the mass media about its high infectivity, mortality, diseases predisposing to the adverse course of SARS-CoV-2 infection, as well as social isolation, lack of targeted treatment, and limited access to medical care produce a huge mental burden which results in psychological distress and sleep disorders [9,10,11,12,13]. Patients with chronic conditions experience higher stress levels due to being at higher risk for worse outcome from COVID-19 [14]. In an analysis conducted by Addis et al., it was noted that older age, female sex, longer duration of illness, presence of respiratory symptoms and no social support were significantly associated with an abnormal psychological impact of COVID-19 on patients with chronic disease [15]. However, the analyses of mental health disorders refer mainly to health professionals, and rarely to people with chronic diseases [16]. To the best of our knowledge, until now, there has been no study on the psychological impact of COVID-19 among patients with chronic disease in Poland. Therefore, this study represents the first psychological impact of COVID-19 among high-risk groups, chronic disease patients in the West Pomerania region in Poland. Thus, this study aimed to assess the symptoms of insomnia, depression and anxiety, among people with chronic diseases during the COVID-19 pandemic in Zachodniopomorskie province in Poland. The result of our study may help the governmental and medical community in formulating comprehensive interventions to prevent psychological problems of chronic disease clients related to COVID-19.

## 2. Materials and Methods

We conducted a cross-sectional survey among the inhabitants with chronic diseases who attended the inpatient units and outpatient clinics of the West Pomerania region in Poland from 3 May to 17 May 2020. During this period at each inpatient units and outpatient clinics, all consecutive patients who corresponded to the inclusion criteria were invited to complete a 9-item Patient Health Questionnaire (PHQ-9), 7-item Insomnia Severity Index (ISI) and a 7-item Generalized Anxiety Disorder scale (GAD-7) in Polish followed by an interview with a structured socio-demographic questionnaire. The criteria for inclusion were as follows: (1) informed consent before the survey was conducted; (2) residence in the West Pomerania region; (3) age 18 years and older. Exclusion criteria: diagnosis of mental illness. Each participant provided details regarding their basic demographic data and chronic diseases such as hypertension, diabetes mellitus, heart failure, coronary heart disease, dyslipidemia, chronic obstructive pulmonary disease and nicotinism. Each patient was also asked about the presence of Hashimoto’s disease, one of the most common, if not the most common, autoimmune disease. The survey involved 879 people. All participants gave their informed consent before the survey was conducted. These participants could interrupt the survey at any time. The full confidentiality of information was ensured. Prior to the survey, the opinion of the Bioethics Committee of the Pomeranian Medical University in Szczecin was received (KB-0012/26/04/2020/Z) which conformed to the ethical guidelines of the Declaration of Helsinki.

Insomnia symptoms were assessed using the 7-item Insomnia Severity Index (ISI; range 0–28) [17,18,19], depression symptoms using the 9-item Patient Health Questionnaire (PHQ-9; range 0–27) [20,21,22], and anxiety symptoms with the 7-item Generalized Anxiety Disorder scale (GAD-7; range 0–21) [23]. 

### Statistical Analysis

Statistical analysis was conducted using Statistica 13.0 software (StatSoft, Tulsa, OK, USA). The Shapiro-Wilk test was used to assess the distribution of data. Mann–Whitney’s U test was performed to analyze quantitative. *X*^2^ test was applied for qualitative data; the Yates correction was used if the subgroup size was insufficient. Correlation analysis was performed using the Spearman’s rank correlation coefficient. The multivariable logistic regression model analysis was used to evaluate the relationship between the analyzed parameters. It was corrected for potentially distorting data (gender, age, dyslipidemia, diagnosed hypertension, and cigarette smoking). Differences were deemed statistically significant at *p* ≤ 0.05. 

## 3. Results

### 3.1. Evaluation of Mean Scores on Generalized Anxiety Disorder (GAD-7), Patient Health Questionnaire (PHQ-9), and Insomnia Severity Index (ISI) Scales

Female subjects significantly more often presented symptoms of anxiety, depression and insomnia. Differences in mean scores on GAD-7 (*p* = 0.004), PHQ-9 (*p* = 0.013) and ISI (*p* = 0.006) were observed between patients with and without hypertension. Moreover, statistically significant differences in mean GAD-7, PHQ-9 and ISI scores were found between smoking and non-smoking patients (*p* < 0.001) and between patients with and without Hashimoto (*p* < 0.001). In addition, significant differences were found between patients with and without dyslipidemia in mean ISI scores (*p* = 0.035). The mean GAD-7, PHQ-9, and ISI scores are presented in Table 1. 

### 3.2. Analysis of Correlation between Selected Parameters and GAD-7, PHQ-9, and ISI Scores

The correlation analysis showed a strong correlation between the occurrence of Hashimoto’s disease and an increase in ISI score (r = 0.797, *p* < 0.001), GAD-7 scale (r = 0.766, *p* < 0.001) and PHQ-9 scale (r = 0.767, *p* < 0.001). A case comparison is presented in Table 2. Therefore, for further analysis, only Hashimoto’s disease was included as a predictor of insomnia, anxiety, or depression.

### 3.3. Evaluation of Selected Parameters Depending on the Occurrence of Hashimoto’s Disease

The analysis of the selected parameters depending showed statistically significant differences in gender (*p* < 0.001), age (*p* < 0.001), hypertension (*p* < 0.001), dyslipidemia (*p* = 0.004) and smoking cigarettes (*p* < 0.001) between participants with and without Hashimoto. The results are presented in Table 3. 

### 3.4. Mental Health and Insomnia in Patients with Hashimoto’s Disease

After the results were corrected for age, gender, diagnosed hypertension, dyslipidemia and cigarette smoking, it was confirmed that the diagnosis of Hashimoto’s disease was associated with an increased risk of anxiety (GAD-7, OR = 2.225; *p* < 0.001), depression (PHQ-9, OR = 2.518; *p* < 0.001) and insomnia (ISI, OR = 3.530; *p* < 0.001). The results are presented in Table 4. 

## 4. Discussion

The COVID-19 pandemic has affected many aspects of human lives worldwide. Information about the diseases that predispose to severe and unfavorable course of COVID-19, the spectre of long-term quarantines, lack of targeted treatment or lack of medical support have been causing a huge psychological burden and generating psychological distress and sleep disorders.

Research on psychological distress and sleep disorders in patients with chronic diseases shows that they are particularly predisposed to mental health disorders [24]. The reported incidence of depression, anxiety or sleep disorders varies widely, depending on the population studied and diagnostic tools. In an analysis conducted by Pumar et al., it was noted that in patients with chronic obstructive pulmonary disease the prevalence of depression ranges from 10% to 57% and in the case of anxiety the prevalence ranges from 7% to 50% [25]. Budhiraja et al. noted sleep disorders in 27.3% of patients with chronic obstructive pulmonary diseases [26]. Other studies have confirmed the relationship between depression, anxiety and sleep disorders in patients with chronic pain, chronic neurological diseases, kidney diseases or autoimmune diseases [27,28,29,30,31].

The most important finding of the present study is the fact that among chronic diseases such as hypertension, diabetes, coronary artery disease, circulatory failure, dyslipidemia, chronic obstructive pulmonary disease, nicotinism, and Hashimoto’s disease, it was the group of patients with Hashimoto’s disease that showed a strong correlation with increased scores on ISI (r = 0.797, *p* < 0.001), GAD-7 (r = 0.766, *p* < 0.001) and PHQ-9 scales (r = 0.767, *p* < 0.001). The group of patients with Hashimoto’s disease also showed significantly more frequent symptoms of anxiety, depression and insomnia compared to people without this condition (*p* < 0.001, *p* < 0.001, *p* < 0.001). 

These individuals, also after the results were corrected for age, gender, hypertension, dyslipidemia and smoking, showed more than twice higher risk of aggravation of anxiety symptoms, more than 2.5 times greater increase in the severity of depression symptoms, and more than 3.5 times increase in the severity of insomnia. 

Autoimmune thyroiditis often coexists with insomnia, depression and anxiety [32]. The prevalence of antithyroid autoantibodies in patients with depressive disorders is higher than in the general population. Carta et al. showed that the risk of depressive disorders in a group of patients with thyroid diseases was up to six times higher than in those without, regardless of thyroid dysfunction assessed by routine serological tests [33]. Similar results were obtained by Giynas et al. who confirmed an increased incidence of depressive disorders in a group of patients with thyroid diseases [34]. Kirim et al. pointed out an increased risk of depression in a group of patients with chronic inflammation of the thyroid gland even though its hormonal function was normal. In addition, they showed that an autoimmune inflammation of the thyroid gland can result in the failure of the thyroid gland, which in turn is a risk factor for the development of depression resistant to standard treatment [35]. Geracioti et al. described an interesting case of a patient with classic symptoms of emotionally unstable borderline personality, with co-occurring autoimmune thyroiditis, in whom mood swings and psychotic symptoms were directly related to the titer of antithyroid antibodies [36]. In a study by Huang et al., the severity of depression and insomnia were significantly correlated with low FT3 [37]. In anxiety disorders, the first symptom in patients with hypothyroidism is often generalized agitation [38]. Since this disorder commonly coexists with elevated blood pressure and tachycardia, it can be assumed that this condition leads to the development of generalized anxiety syndrome.

Suffering from an autoimmune disease can generate enormous stress, through a significant reduction in activity at home or at work, financial difficulties related to the cost of medical care and reduced income, lack of acceptance of its appearance resulting from, among other things, complications of the applied treatment, impaired interpersonal relations or loss of independence [39]. Even discreet intensification of everyday stress factors in people with autoimmune diseases affects the hypothalamic-pituitary-adrenal axis homeostasis, which leads to intensification of the disease symptoms or adversely affects its remission [40]. It seems that is why patients with Hashimoto’s disease, who are aware of an increased risk of more severe symptoms of COVID-19, are most likely to develop anxiety, depression and insomnia among all chronic diseases, as confirmed by our study. 

Our study had several limitations. Firstly, Hashimoto’s disease was the only autoimmune disease that was included in this study, although there may be other autoimmune diseases which may induce similar psychological effects during the SARS-CoV-2 pandemic. Secondly, there is an interplay between depressive and anxiety symptoms on the one hand and the presence of insomnia symptoms on the other hand. Thirdly, as a cross-sectional study, it provided no information on any change in mental health of the respondents.

## 5. Conclusions

This study showed that patients with a diagnosis of Hashimoto’s disease during the COVID-19 pandemic are at a higher risk of insomnia, depression and anxiety than patients with other chronic diseases. Therefore, people with Hashimoto’s disease during the pandemic require appropriate medical support, information, stress reduction and rest.

The result of our study may help the governmental and medical community in formulating comprehensive interventions to prevent psychological problems among chronic disease clients related to COVID-19.

## Figures and Tables

**Table 1 jcm-10-01206-t001:** Mean GAD-7, PHQ-9, and ISI scores for selected parameters.

		GAD-7		PHQ-9		ISI	
		mean ± SD; Me	*p*	mean ± SD; Me	*p*	mean ± SD; Me	*p*
Gender	female	12.49 ± 5.89; 14.0	<0.001	14.25 ± 5.79; 16.0	<0.001	16.64 ± 6.47; 19.0	<0.001
	male	7.61 ± 5.28; 7.0		9.74 ± 5.19; 10.0		11.42 ± 6.15; 12.0	
Hypertension		9.32 ± 5.61; 9.0	0.004	11.51 ± 5.45; 11.0	0.013	13.55 ± 5.70; 13.0	0.006
Diabetes mellitus		10.00 ± 5.16; 11.0	0.580	11.80 ± 4.61; 10.0	0.392	13.95 ± 4.86; 13.0	0.384
Coronary heart disease		14.50 ± 4.95; 14.5	0.401	17.00 ± 4.24; 17.0	0.300	19.00 ± 2.83; 19.0	0.396
Heart failure		7.00 ± 2.83; 7.0	0.395	9.00 ± 4.24; 9.0	0.389	13.00 ± 2.83; 13.0	0.607
Dyslipidemia		10.12 ± 5.73; 11.0	0.157	12.08 ± 5.73; 12.0	0.142	13.89 ± 6.27; 15.0	0.035
Chronic obstructive pulmonary disease		8.67 ± 4.93; 11.0	0.568	9.00 ± 3.61; 10.00	0.280	9.33 ± 2.08; 10.0	0.126
Smoke cigarettes		7.27 ± 4.74; 7.0	<0.001	9.33 ± 5.01; 10.0	<0.001	11.31 ± 5.75; 13.0	<0.001
Hashimoto’s disease		15.46 ± 2.33; 16.0	<0.001	17.34 ± 2.10; 17.0	<0.001	20.75 ± 2.19; 21.0	<0.001

Abbreviations: *p*–statistical significance; Me–median; SD–standard deviation; GAD-7–Generalized Anxiety Disorder scale; PHQ-9–Patient Health Questionnaire; ISI–Insomnia Severity Index.

**Table 2 jcm-10-01206-t002:** Analysis of correlations between selected parameters and GAD-7, PHQ-9, and ISI scores.

	GAD-7		PHQ-9		ISI	
	R	*p*	R	*p*	R	*p*
Age (years)	−0.082	0.009	−0.082	0.009	−0.077	0.015
Hypertension	−0.092	0.004	−0.079	0.013	−0.087	0.006
Diabetes mellitus	−0.018	0.579	−0.027	0.391	−0.028	0.384
Coronary heart disease	0.027	0.399	0.033	0.299	0.027	0.395
Heart failure	−0.027	0.394	−0.027	0.388	−0.016	0.606
Dyslipidemia	−0.045	0.156	−0.047	0.141	−0.067	0.035
Chronic obstructive pulmonary disease	−0.018	0.567	−0.034	0.279	−0.049	0.125
Smoke cigarettes	−0.350	<0.001	−0.347	<0.001	−0.327	<0.001
Hashimoto’s disease	0.766	<0.001	0.767	<0.001	0.797	<0.001

Abbreviations: *p*–statistical significance; R–correlation coefficient; GAD-7–Generalized Anxiety Disorder scale; PHQ-9–Patient Health Questionnaire; ISI–Insomnia Severity Index.

**Table 3 jcm-10-01206-t003:** Evaluation of selected parameters depending on the occurrence of Hashimoto’s disease.

		without Hashimoto (*n* = 589)	with Hashimoto (*n* = 290)	*p*
Gender (*n*, %)	female	275 (46.69%)	253 (87.24%)	<0.001
	male	314 (53.31%)	37 (12.76%)	
Age (years), mean ± SD; Me		39.71 ± 7.07; 39.0	37.97 ± 5.68; 37.0	<0.001
Hypertension (*n*, %)	no	487 (82.68%)	265 (91.38%)	<0.001
	yes	102 (17.32%)	25 (8.62%)	
Diabetes mellitus (*n*, %)	no	574 (97.45%)	286 (98.62%)	0.383
	yes	15 (2.55%)	4 (1.38%)	
Coronary heart disease (*n*, %)	no	588 (99.83%)	290 (100.00%)	0.717
	yes	1 (0.17%)	0 (0.00%)	
Heart failure (*n*, %)	no	587 (99.66%)	290 (100.00%)	0.809
	yes	2 (0.34%)	0 (0.00%)	
Dyslipidemia (*n*, %)	no	470 (79.80%)	255 (87.93%)	0.004
	yes	119 (20.20%)	35 (12.07%)	
Chronic obstructive pulmonary disease (*n*, %)	no	586 (99.49%)	290 (100.00%)	0.547
	yes	3 (0.51%)	0 (0.00%)	
Smoke cigarettes (*n*, %)	no	329 (55.86%)	277 (95.52%)	<0.001
	yes	260 (44.14%)	13 (4.48%)	

Abbreviations: *p*—statistical significance; *n*—number of patients; % percentage of patients.

**Table 4 jcm-10-01206-t004:** Severity of mental health and insomnia in patients with Hashimoto’s disease (multivariable logistic regression).

Hashimoto’s Disease (Adjusted by Potentially Distorting)				
	*p*	OR	CI −95%	CI+ 95%
GAD-7	<0.001	2.225	1.944	2.546
PHQ-9	<0.001	2.518	2.154	2.943
ISI	<0.001	3.530	2.721	4.581

Abbreviations: *p*–statistical significance; OR- odds ratio; CI-confidence interval; GAD-7–Generalized Anxiety Disorder scale; PHQ-9–Patient Health Questionnaire; ISI–Insomnia Severity Index. Notes: Potentially distorting data (age, gender, diagnosed hypertension, dyslipidemia and cigarette smoking).

## Data Availability

All data that support the findings of this study are available upon request from the corresponding author.

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
