# Peer review of "The Impact of the COVID-19 Pandemic on Psychological Health and Insomnia among People with Chronic Diseases"

_jcm, 2021, doi:10.3390/jcm10061206_

Round 1
Reviewer 1 Report
Dear authors,
I have read with interest your research paper and if overall the project is very interesting, several questions remain:
The first question is why this interest in Hashimoto disease? Is it more prevalent in your region?
Regarding the recruitment methodology, it is not very clear for me how the subjects were selected. Can you please detail this aspect? I think that a chart describing the recruitment and timing would be very helpful
You state that the aim is to investigate the symptoms of sleep disorders, but only an insomnia questionnaire is addressed. Can you please explain? Alternatively, just specify in the results that you speak about insomnia, and not sleep disorders in general.
You evaluated the first three weeks of lockdown, but GAD-7 is evaluating the last 14 days. Were they contacted at the end of this three weeks period or during this period?
Results section
It seems obvious that female subjects presenting anxiety and/or depressive symptoms have as well sleep initiating or maintaining difficulties. Could the presence of pschiatrc symptoms bias the sleep related symptoms?
Hashimoto per se is associated with anxiety, depression and subsequent (or independent) insomnia symptoms. Why do you think is lockdown related?
Prevalence of Hashimoto in your sample seems very high. Can you, please, explain?
For the tables, please add an explanatory legend.
Discussion and limitations
Lines 159-164 maybe not highly appropriated for discussion. This short description of Hashimoto maybe more suitable for introduction.
Lines 146-147: is speaking only about insomnia, not sleep disorders
The diagnosis was retained i think exclusively on patient description, but we have no information on the actual status of their comorbidities (controlled/treated or not). Furthermore, we can not see if the factors you evaluated were present before the lockdown or if their severity altered during this context (you speak about the aggravation of symptoms on lines 156-158).
Many thanks
Author Response
Response to Reviewer 1 Comments
Dear Reviewer,
Thanks a lot for your review and valuable tips. Answers to your questions are down below:
Point 1: The first question is why this interest in Hashimoto disease? Is it more prevalent in your region?
Response 1: Due to the fact that on a daily basis I look after patients with neuropsychiatric complications in various autoimmune diseases, I pay special attention to this group of patients. Of all autoimmune diseases, Hashimoto's disease is the most common autoimmune entity. This is also true of our study in which a significant proportion of patients suffer from Hashimoto's disease. Another issue is that I am trying to reach the awareness of doctors of other specialties, emphasizing the seriousness of autoimmune diseases in our society. Currently, Hashimoto's disease is rarely given special attention, unfortunately often disregarding patients with this diagnosis.
Point 2: Regarding the recruitment methodology, it is not very clear for me how the subjects were selected. Can you please detail this aspect? I think that a chart describing the recruitment and timing would be very helpful
Response 2: According to the reviewer's instructions, we have completely modified the materials and methods section to make it understandable for all readers.
Point 3: You state that the aim is to investigate the symptoms of sleep disorders, but only an insomnia questionnaire is addressed. Can you please explain? Alternatively, just specify in the results that you speak about insomnia, and not sleep disorders in general.
Response 3: Thank you very much for your valuable tip. We changed the terminology in the text from sleep disorder to insomnia, in line with the reviewer's recommendation so as not to mislead the reader.
Point 4: You evaluated the first three weeks of lockdown, but GAD-7 is evaluating the last 14 days. Were they contacted at the end of this three weeks period or during this period?
Response 4: In Poland, the epidemic state was introduced on 20 March 2020. However, our study was conducted from May 3 to May 17, 2020. Each of the study participants commented on the period of the last 14 days, i.e. the so-called closing period.
Point 5: It seems obvious that female subjects presenting anxiety and/or depressive symptoms have as well sleep initiating or maintaining difficulties. Could the presence of pschiatrc symptoms bias the sleep related symptoms?
Response 5: Dear reviewer thank you for this question. From my experience, I can say that yes, psychiatric symptoms can induce or worsen symptoms of sleep disorders. However, we did not include patients with psychiatric diseases in our study, so I cannot draw such conclusions in relation to our study.
Point 6: Hashimoto per se is associated with anxiety, depression and subsequent (or independent) insomnia symptoms. Why do you think is lockdown related?
Response 6: Even discreet intensification of everyday stress factors in people with autoimmune diseases affects the hypothalamic-pituitary-adrenal axis homeostasis, which may leads to the onset of neuropsychiatric symptoms intensification of the disease symptoms or adversely affects the time of its remission
Point 7: Prevalence of Hashimoto in your sample seems very high. Can you, please, explain?
Response 7: Due to the fact that on a daily basis I look after patients with neuropsychiatric complications in various autoimmune diseases, I pay special attention to this group of patients. Of all autoimmune diseases, Hashimoto's disease is the most common autoimmune entity. This is also true of our study in which a significant proportion of patients suffer from Hashimoto's disease
Point 8: For the tables, please add an explanatory legend.
Response 8: As recommended by the reviewer, we have added a legend to the tables
Point 9: Lines 159-164 maybe not highly appropriated for discussion. This short description of Hashimoto maybe more suitable for introduction.
Response 9: Following the reviewer's comment, we have moved the Hashimoto short description from discussion to introduction. Line 56-61.
Point 10: Lines 146-147: is speaking only about insomnia, not sleep disorders
Response 10: We changed the terminology in the text from sleep disorder to insomnia, in line with the reviewer's recommendation so as not to mislead the reader.
Point 11: The diagnosis was retained i think exclusively on patient description, but we have no information on the actual status of their comorbidities (controlled/treated or not). Furthermore, we can not see if the factors you evaluated were present before the lockdown or if their severity altered during this context (you speak about the aggravation of symptoms on lines 156-158).
Response 11: Exactly, we had no way of judging whether comorbidities were present before lockdown. We collected information on chronic diseases using a binary system. On the other hand, our exclusion criterion was mental illness, so the people included in the study did not suffer from mental disorders prior to inclusion in our study.
Best regards
PhD Paweł Wańkowicz
Reviewer 2 Report
The authors need to expand the review of literature that is relevant to their study. I encaourage to include and discuss the following articles in the "Introduction" section
Addis, S. G., Nega, A. D., & Miretu, D. G. (2021). Psychological impact of COVID-19 pandemic on chronic disease patients in Dessie town government and private hospitals, Northeast Ethiopia. Diabetes & Metabolic Syndrome: Clinical Research & Reviews, 15(1), 129-135.
Kendzerska, T., Zhu, D. T., Gershon, A. S., Edwards, J. D., Peixoto, C., Robillard, R., & Kendall, C. E. (2021). The Effects of the Health System Response to the COVID-19 Pandemic on Chronic Disease Management: A Narrative Review. Risk management and healthcare policy, 14, 575.
The goal of the study needs to be properly highlighted and justified. Instead of setting their aim in the frame of a simple question, I would recommend that the authors attempt to present the key objectives of their study with regards to what is presently known (i.e. literature), thus highlighting the added value of the article.
The authors are encouraged to add a Procedure section and describes HOW the experiment was done and how the data was collected.
The authors should discuss why (from both theoretical and practical perspectives) why the group of patients with Hashimoto's disease showed a strong correlation with some indicators such as increased scores on ISI, GAD-7 and PHQ-9 scales
Author Response
Response to Reviewer 2 Comments
Dear Reviewer,
Thanks a lot for your review and valuable tips. Answers to your questions are down below:
Point 1: The authors need to expand the review of literature that is relevant to their study. I encaourage to include and discuss the following articles in the "Introduction" section
Addis, S. G., Nega, A. D., & Miretu, D. G. (2021). Psychological impact of COVID-19 pandemic on chronic disease patients in Dessie town government and private hospitals, Northeast Ethiopia. Diabetes & Metabolic Syndrome: Clinical Research & Reviews, 15(1), 129-135.
Kendzerska, T., Zhu, D. T., Gershon, A. S., Edwards, J. D., Peixoto, C., Robillard, R., & Kendall, C. E. (2021). The Effects of the Health System Response to the COVID-19 Pandemic on Chronic Disease Management: A Narrative Review. Risk management and healthcare policy, 14, 575.
Response 1: We have modified the introduction section according to the reviewer's recommendations and guidelines within the limits that do not change the narrative of the manuscript.
Point 2: The goal of the study needs to be properly highlighted and justified. Instead of setting their aim in the frame of a simple question, I would recommend that the authors attempt to present the key objectives of their study with regards to what is presently known (i.e. literature), thus highlighting the added value of the article.
Response 2: In accordance with the recommendation of the reviewer we highlighted the aim of our study. Line 80-87.
Point 3: The authors are encouraged to add a Procedure section and describes HOW the experiment was done and how the data was collected.
Response 3: In accordance with the reviewer's instructions, we have completely modified the material and methods section to make it transparent and obvious to the reviewer and readers. Line 90-100.
Point 4: The authors should discuss why (from both theoretical and practical perspectives) why the group of patients with Hashimoto's disease showed a strong correlation with some indicators such as increased scores on ISI, GAD-7 and PHQ-9 scales
Response 4: Following the reviewer's instructions, we have expanded the discussion. Line 193-207.
Best regards
PhD Paweł Wańkowicz
Round 2
Reviewer 1 Report
Dear authors,
Thank you for your reponse. It is much clearer for me now in terms of study design.
Concerning the limitations I would delete the sample size limitation, because the sample is sufficient for this purpose. I would add instead the interplay between depressive and anxiety symptoms on one hand and the presence of insomnia symptoms on the other hand.
Author Response
Dear Reviewer,
Thanks a lot for your review and valuable tips.
Point 1: Concerning the limitations I would delete the sample size limitation, because the sample is sufficient for this purpose. I would add instead the interplay between depressive and anxiety symptoms on one hand and the presence of insomnia symptoms on the other hand.
Response 1: According to the reviewer's instructions, we have modified the limitation. Line 402-403.
Best regards
PhD Paweł Wańkowicz
Reviewer 2 Report
The paper is now suitable for publication
Author Response
Dear Reviewer,
Thanks a lot for your review.
Best regards
PhD Paweł Wańkowicz
This manuscript is a resubmission of an earlier submission. The following is a list of the peer review reports and author responses from that submission.